# The Cost Benefit of Refinery Effluent Pretreatment Upstream of Membrane Bioreactors

**DOI:** 10.3390/membranes13080715

**Published:** 2023-08-01

**Authors:** Kasro Kakil Hassan Dizayee, Adil Mohammed Raheem, Simon J. Judd

**Affiliations:** 1College of Engineering, Salahaddin University-Erbil, Erbil 44002, Iraq; kasro.dizayee@su.edu.krd; 2College of Engineering, Al-Kitab University, Altun Kupri, Kirkuk 36001, Iraq; adil.m.raheem@uoalkitab.edu.iq; 3Cranfield Water Science Institute, Cranfield University, Bedford MK43 0AL, UK

**Keywords:** API separators, cost, dissolved air flotation, energy consumption, membrane bioreactors, refinery effluent

## Abstract

The established classical method of treating oil refinery effluent is flotation followed by biological treatment. Membrane bioreactors (MBRs) offer more advanced treatment, producing a clarified and potentially reusable treated effluent, but demand robust pretreatment to remove oil and grease (O&G) down to consistent, reliably low levels. An analysis of a full-scale conventional oil refinery ETP (effluent treatment plant) based on flotation alone, coupled with projected performance, energy consumption and costs associated with a downstream MBR, have demonstrated satisfactory performance of flotation-based pretreatment. The flotation processes, comprising an API (American Petroleum Institute) separator followed by dissolved air flotation (DAF), provided ~90% removal of both total suspended solids (TSS) and O&G coupled with 75% COD (chemical oxygen demand) removal. The relative energy consumption and cost of the pretreatment, normalised against both the volume treated and COD removed, was considerably less for the API-DAF sequence compared to the MBR. The combined flotation specific energy consumption in kWh was found to be almost an order of magnitude lower than for the MBR (0.091 vs. 0.86 kWh per m^3^ effluent treated), and the total cost (in terms of the net present value) around one sixth that of the MBR. However, the nature of the respective waste streams generated and the end disposal of waste solids differ significantly between the pretreatment and MBR stages.

## 1. Introduction

The treatment of effluents generated by the petroleum industry poses a number of challenges for treatment, both generally [1] and by membrane bioreactor (MBR) technology specifically [2,3]. Oil refinery effluents comprise an aggregation of a number of different wastewater streams generated by the refining of crude oil (or “crude”) by thermal fractionation to generate useful mineral oil-based products [3]. These streams significantly differ in their composition, and are discharged at different times over the various refining sub-process cycles, leading to significant temporal variations in the aggregated effluent quality. Partly as a consequence of this, reported refinery effluent composition from different studies vary widely.

Studies of the application/implementation of MBR technology for refinery effluent treatment have tended to appraise biological treatment performance in terms of the removal of COD and oil factions at both bench-scale [4,5,6,7,8,9,10] and pilot/full-scale [11,12,13,14]. Most of these studies also provide the membrane flux and/or permeability values, and these appear to vary widely. A recent review of full-scale MBR plants [3] reported mean operating flux values of 9–15 L·m^−2^·h^−1^ (LMH) at mean influent COD levels between 150 and 990 mg·L^−1^, with no evident correlation between the flux and COD load.

Two key specific and widely recognized challenges to biological treatment of refinery effluents by MBR technology are:(a)The fluctuation in salt concentration, due to intermittent discharges from the desalter;(b)The presence of suspended (or “free”) oil and grease (O&G).


Salinity shocks can perturb the microbial ecology in the biotreatment process, causing the release of extracellular polymeric substances (EPS), which are membrane foulants and can also promote foaming. High salinity can further inhibit the essential process biochemistry. O&G can accumulate on the screens and within the biological process tank and increase the hydrophobicity of the mixed liquor, resulting in a significant reduction in the membrane permeability [6,8]. While the challenge of fluctuating salinity loads can be addressed by using flow equalisation, O&G must be substantially removed prior to the MBR step—normally through flotation (Figure 1). Flotation usually comprises both gravity separation, using a simple API (American Petroleum Institute) separator followed by an air- or gas-assisted separation process of induced gas or dissolved air flotation (IGF or DAF, respectively).

Both published research at the bench scale [4,5,6,7,8] and performance measurements pertaining to pilot and full-scale plants [3,11,14] have demonstrated reasonably effective removal of O&G by MBR technology. Removals of both O&G and COD have tended to exceed 90% according to bench-scale studies conducted under controlled conditions (Table 1), but apparently decrease at high salinity levels [7]. The data in Table 1 indicates there to be some sensitivity of the fouling propensity to the combination of flux and O&G levels. Fouling appears to be largely mitigated by operation at unrealistically low flux levels, allowing extremely high feed O&G concentrations [5,7]. At more practical flux values, evidence suggests that the O&G MBR feed concentration must be kept below 100 mg/L [6,12].

Notwithstanding correlations produced at bench-scale, influent O&G concentration limits are recommended by the MBR membrane technology suppliers based on operational knowledge and experience. These limits may apply to either the biological stage or the membrane separation stage. At the biological stage, threshold concentrations of 50 mg/L and 150 mg/L have been recommended by the suppliers Toray and Microdyn Nadir, respectively, for fats oils and grease (or FOG), whereas at the subsequent membrane separation stage, the suggested limit is considerably more conservative (10 mg/L, according to Microdyn Nadir). Against this, the removal efficiency appears to be fairly robust to changes in organic load, as represented by the COD or BOD concentration, according to bench-scale studies [4] and data from full-scale refinery and petrochemical plants [2,14].

Whilst there have been a few studies establishing the efficacy of MBR technology for providing enhanced biological treatment of refinery effluents, the precise requirements and possible technoeconomic impacts of the pretreatment have been largely overlooked. Studies of gas or air-assisted flotation have tended to focus on the scientific aspects of the process [15,16], and those dedicated to refinery effluent treatment have been based predominantly in coagulation optimisation [17,18]. Few studies appear to have considered key engineering process facets—such as the cost benefit offered—within the context of refinery effluent treatment. The current study aims to examine these largely unexplored aspects with reference to a full-scale refinery effluent treatment plant (ETP), the specific objectives being to:Establish the treated effluent COD and O&G concentrations attained by the conventional physicochemical, flotation-based ETP;Quantify the energy requirements (as the specific energy consumption, SEC), chemical demand, and capital and operating expenditure (CAPEX and OPEX) associated with the above;Compare these key parameter values with those projected for downstream treatment by an MBR.


## 2. Method

### 2.1. Reference Site

The reference site used for the study was the Kawrgosk Oil Refinery, situated 28 km from the city of Erbil in the Kurdistan region of Northern Iraq. The Kawrgosk Effluent Treatment Plant (KETP) process treatment train comprises the classical three-stage process for the physicochemical separation (Figure 1), with supplementary steps of:(a)A preliminary separation step for bulk separation of the suspended oil;(b)An additional neutralisation (pH adjustment) tank fitted between the API unit and the equalisation (EQ) tank.


Residence times have been calculated from the individual unit process dimensions for an assumed flow rate of 75 m^3^·h^−1^ (Table 2). The performance of each individual unit process step was assessed through measurement of the COD, O&G, and TSS (total suspended solids) concentrations along with the turbidity upstream and downstream of the API separator and DAF. The DAF received a dose of 25–40 mg/L polyaluminium chloride along with polyacrylamide flocculant. A total of 16 sets of samples were taken between March and June 2022. Standard methods [19] were used for all wastewater quality determinations.

### 2.2. Cost Analysis

The total cost was determined as the net present value (NPV) expressed in terms of the CAPEX and OPEX [20]:(1)$$\mathrm{NPV}=\sum_{t=0}^{t=t}\frac{{\mathrm{CAPEX}}_{t=0}+{\mathrm{OPEX}}_{t}\;}{{\left(1+D\right)}^{t}}$$
where D is the discount factor (assumed to be 5%) and *n* is the total plant life (or amortisation period), taken as 30 years.

The normalised CAPEX value (*L_C_* = CAPEX per unit flow rate, USD per m^3^/d) of the MBR was determined using the method of Jalab et al. [21]. These authors captured published CAPEX data (Table 3) either from specific existing installations [22,23] or commercial CAD software [24], to generate cost curves (*L_C_* as a function of flow capacity *Q_P_*). Further economic assessments have been conducted in which the cost of individual components has been collated and the total cost determined through aggregation [25,26]. The latter have tended to underestimate the actual cost, since the nominal cost of assembly along with the marginal costs (profit, contingency, etc.) are not readily estimated. As with the previous study [21], the CAPEX information was restricted to the cost of the installed capital equipment, excluding costs specific to the locality of the installation (i.e., civil engineering costs).

The normalised OPEX values (*L_O_* in USD·m^−3^) were calculated from energy and consumables consumption. For all pretreatment processes other than the DAF and API, the energy costs were assumed to be negligible. Baseline values for chemical consumption and waste generation were provided by the Kawrgosk Oil Refinery owners. Previously published OPEX values for the MBR [21] were updated for changes in commodity prices, and specifically energy, chemicals and membranes (Table 4), according to the Chemical Engineering Plant Cost Index (CEPCI), and used to determine the total normalised OPEX in USD/m^3^ according to the method of Jalab et al. [21] (Table 5). Waste disposal costs were excluded since the cost of arguably the most sustainable option of mono-incineration varies considerably [28]. CAPEX data for the flotation processes were taken from textbook sources [29].

The total energy consumption for the MBR is the sum of the process biology aeration demand *E_A,bio_*, and the energy associated with membrane permeation *E_m_*. The latter is the sum of the air scour energy, the energy for sludge recirculation between the membrane and process tank, and the membrane permeation energy. The cost associated with this energy consumption is added to the other non-energy related cost components (membrane replacement, chemicals consumption and waste disposal) to give the total OPEX.

Annualising all scheduled OPEX at a discount factor of 5% over a 30 yr amortisation period simplifies Equation (1) to:(2)$$\mathrm{NPV}=Q_{P}L_{C}+365Q_{P}L_{O}\sum_{t=0}^{t=t}\frac{1\;}{{\left(1+D\right)}^{t}}=Q_{P}(L_{C}+5976L_{O})$$

The above equation was used to compare the total cost incurred by the individual key process steps of the envisaged treatment train, namely the API, DAF and MBR. The operating cost *L_O_* for the MBR includes membrane replacement every 8 years.

## 3. Results and Discussion

### 3.1. Pretreatment Unit Process Performance

The pollutant load in kg/h to the individual process steps varied as a function of both pollutant concentration and flow rate (55–75 m^3^/h). Correlations of the outlet and inlet TSS, O&G and COD loads for the DAF (Figure 2) revealed a near linear relationship (R^2^ = 0.980–0983) for all three parameters, with a lower mean percentage removal for the COD (69%) than for the TSS and O&G (85–86% for both) due presumably to the substantial soluble fraction. The outlet concentrations for these three water quality determinants were thus proportional to the inlet load, and thus subject to a degree of data scatter as indicated by the standard deviation values (Figure 3).

A comparison of the removals attained by the API and DAF stages (Figure 4) indicate the expected significantly greater % removal of all the pollutants by the DAF than by the API. However, contrary to the DAF trend, the API achieved higher COD than O&G and TSS removal (29% cf. 17–18%), suggesting that the COD removed at that stage was largely associated with the TSS and O&G.

Removals of COD by the DAF were significantly lower than those which have been reported previously for pilot-plant trials on oil refinery effluent. O&G, COD and TSS removals of 92–98% have been reported for a DAF using a 100 mg·L^−1^ dose of alum applied under optimal conditions of air saturator pressure and air–water ratios [17]. However, there is clearly an economic incentive for minimising the required coagulant dose. Much lower COD removals of around 68%—comparable with those values employed in the current study—were obtained at a lower coagulant dose of around 25 mg·L^−1^ based on a previous bench-scale test [18]. The maximum coagulant dose applied in the current study was 40 mg/L.

### 3.2. Pretreatment Unit Costs

A consistent basis for comparing the CAPEX of the two physical separation stages is through using textbook information for CAPEX generated on the same basis [29]. Accordingly, a reference value for the CAPEX is provided for a specific flow rate and a correction exponent used to adjust this CAPEX value for different flows or, in the case of the DAF, surface area, viz:*Corrected CAPEX* = *Reference CAPEX* · (*X*_corrected_/*X*_reference_)^*r*^(3)
where *X* refers to flow capacity or surface area. The CAPEX values can then be corrected for 2023 USD using the CEPCI index. The resulting *L_C_* data (Table 6) indicate that, as expected, the DAF incurs a significantly higher capital cost than the API.

The pretreatment OPEX is largely associated with the energy and chemicals consumption. The low-energy and chemical-free nature of the API separator means that this unit operation contributes little to the OPEX. It is therefore reasonable to limit the pretreatment OPEX evaluation to the DAF.

Based on the upper threshold dose of 40 mg/L alum and a bulk cost of USD 350 per ton, the cost of coagulant dosing equates to USD 0.014 m^−3^. The cost of polymer dosing can be conservatively estimated as being similar to this value, giving a total of USD 0.028 m^−3^ for DAF chemical dosing. The SEC for DAF is widely accepted as being in the range 0.05–0.075 kWh·m^−3^ [30], though lower values in the range 0.035–0.047 kWh·m^−3^ have been recently reported [31] and a mean value of 0.05 kWh·m^−3^ determined at a full-scale dairy effluent treatment plant [32]. For the current study, the total installed power of the recirculation pump, compressor, skimmer and mixers was 12.7 kW (Table 2), equating to an SEC of 0.127 kWh·m^−3^ at the maximum flow of 100 m^3^·h^−1^ specified for the DAF. The absorbed power would therefore always be lower than this threshold value.

An SEC of 0.075 kWh·m^−3^ infers an absorbed power of around 60% of the installed power and equates to an energy cost of USD 0.015 m^−3^ for an assumed tariff of USD 0.2 kWh^−1^—the approximate domestic tariff in the US in 2023. The *L_O_* value for the DAF stage is accordingly USD 0.043 m^−3^. For an SEC based on the installed power, the DAF *L_O_* value is USD 0.053 m^−3^.

From Equation (2), the NPV for the API and DAF stages is:$${\mathrm{NPV}}_{\mathrm{API}+\mathrm{DAF}}=Q_{P}(L_{C}+5976L_{O})=1800\left(255+815+5976\cdot L_{O}\right)$$
where *L_O_* ranges between USD 0.043 and USD 0.053 m^−3^. The NPV is thus in the range of USD 2.4–2.5 m, ignoring labour and waste disposal costs. The mean flow of the waste stream generated is 4.2 m^3^/d, constituting ~0.5% of the feed flow at the flow rate of 1800 m^3^·d^−1^.

### 3.3. Projected MBR Costs

A specific CAPEX curve for a municipal MBR, *L_C,MBR_* in kUSD per m^3^·d^−1^ has been presented [21], based on published data sets for municipal wastewater treatment [22,23,24] (Table 3). According to this analysis, *L_C,MBR_* follows a power law relationship with *Q_P_*:*L_C,MBR_* = *m Q_P_^n^*
where the coefficient *m* = 167 and the exponent *n* = −0.462. This yields an *L_C_*_,*MBR*_ of USD 5.23 k per m^3^·d^−1^ flow rate at the flow capacity *Q_P_* of 1800 m^3^·d^−1^, giving a total CAPEX of USD 9.4 m in 2019. Allowing for inflation based on the annual CEPCI values, the projected CAPEX for 2023 for a US-located installation would be USD 12.3 m, hence *L_C_*,*_MBR_* = USD 6870 per m^3^/d, to which the membrane cost contributes ~4.5% based on a specific area cost *L_M_* of USD 85 m^−2^ and an assumed flux value *J* of 12 LMH based on available published data [3].

The values of the individual specific energy consumption (SEC) and OPEX components were calculated from the governing equations (Table 5) to give the total energy consumptions associated with the membrane operation and process biology (Table 7).

The total SEC for the MBR is thus 0.857 kWh·m^−3^. Based on the assumed energy cost of USD 0.2 kWh^−1^, the energy costs for the MBR come to USD 0.171 m^−3^. Added to an estimated chemicals cost of USD 0.02 m^−3^ the total MBR OPEX is USD 0.191 m^−3^.

The NPV of the MBR process, according to Equation (2), is then:$${\mathrm{NPV}}_{\mathrm{MBR}}=Q_{P}(L_{C}+5976L_{O})=1800\left(6870+5976\cdot 0.191\right)=\mathrm{USD}\;14.4\;m$$

It is germane to consider the performance in terms of the SEC and cost normalised against the COD load removed (Table 8). This provides a true indication of the value provided by the individual process steps.

Based on these figures, it is evident that:The DAF removes more than double the COD load of the downstream MBR;The energy consumption of the pretreatment stages is an order of magnitude less than for the MBR;The overall cost, as represented by the NPV, of the pretreatment is around one sixth of that of the MBR step;The NPV normalised against the COD removed is around 20 times less for the pretreatment than for the MBR.

The higher COD load removed by the DAF compared with the MBR is a consequence of the higher influent concentration of the suspended COD for the DAF. This permits efficient removal at a relatively low retention time of ~50 min (Table 2), compared with a projected retention time of around 12 h for the MBR.

The observations regarding energy consumption and relative COD removed corroborates the conclusions from a recent study of dairy effluent treatment based on a comparable treatment scheme [32]. The observation regarding the total cost, as represented by the net present value, reflects both the relatively low cost of the API and DAF units compared with the MBR and the high energy consumption and capital cost associated with biochemical degradation of the COD. Given that the latter is roughly proportional to the COD load, optimising the DAF to achieve higher COD removal will almost certainly provide a cost benefit.

However, the DAF incurs a significant chemical demand and generates a waste stream containing the undegraded organic matter and with a relatively high inorganic content compared with the waste stream from the MBR. The cost of managing this waste stream, and the associated end disposal of the waste solids, thus presents a significant component of the overall cost. Arguably the most sustainable widely implemented method for this duty has been mono-incineration, the cost of which varies widely according to different governing factors [28]. Disposal at the KETP site is currently through landfilling or containerised storage on site—neither of which can be considered sustainable in the long term.

## 4. Conclusions

Practical measurement of the performance of a representative full-scale pretreatment scheme for MBR purification of refinery effluent treatment has demonstrated the API-DAF sequence to be extremely cost effective, according to a novel cost benefit analysis based on the evaluation of the net present value. Around 90% removal of TSS and O&G is achieved by this sequence, with the O&G reliably removed to levels below 50 mg/L in the treated effluent, thereby affording a downstream MBR reasonable protection against membrane fouling. In terms of the COD, the removal is significantly lower (around 75% overall) compared with a projected >90% removal by the MBR. However, the API-DAF specific energy consumption in kWh per m^3^ effluent treated is almost an order of magnitude lower than for the MBR (0.091 vs. 0.86 kWh·m^−3^), and the total cost (in terms of the NPV) around one sixth that of the MBR. Moreover, when normalised against the COD removed, the pretreatment is fifteen times more energy efficient than the MBR (0.066 vs. 1.03 kWh·kgCOD^−1^).

The data indicate that investment in the pretreatment steps to optimize COD removal is likely to provide a cost benefit since the energy demanded by aerobic COD removal by the biological process, such as the MBR, is directly proportional to the COD concentration. However, a key variable is the cost of the management and end disposal of the waste solids. There is a clear need to encompass this element in future cost analyses.

## Figures and Tables

**Figure 1 membranes-13-00715-f001:**
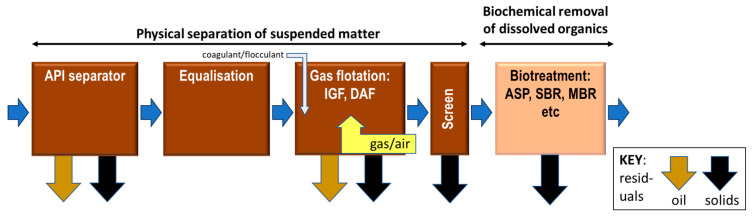
Typical oil refinery effluent treatment scheme, including a biological process step. (IGF—induced gas flotation; DAF—dissolved air flotation; ASP—activated sludge process; SBR—sequencing batch reactor; and MBR—membrane bioreactor).

**Figure 2 membranes-13-00715-f002:**
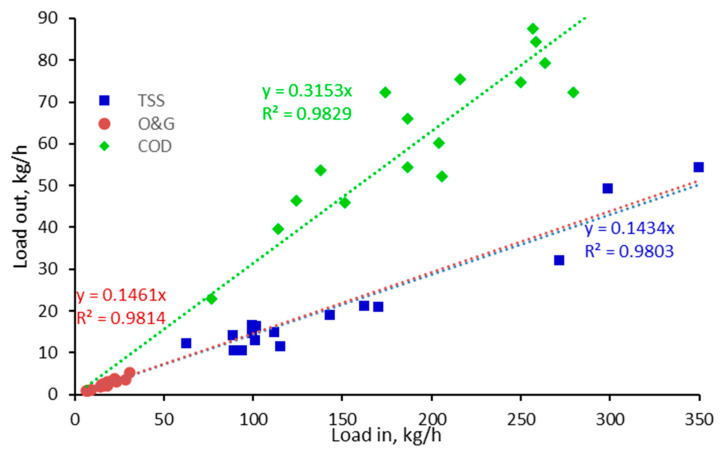
Outlet and inlet load, DAF.

**Figure 3 membranes-13-00715-f003:**
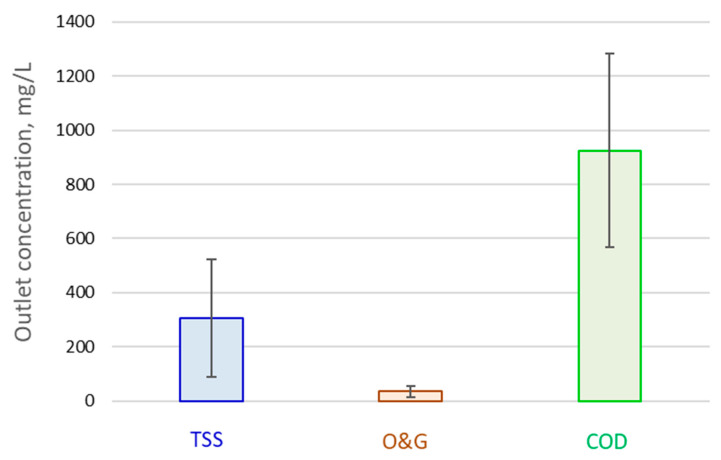
DAF outlet and inlet concentrations, mean and standard deviation.

**Figure 4 membranes-13-00715-f004:**
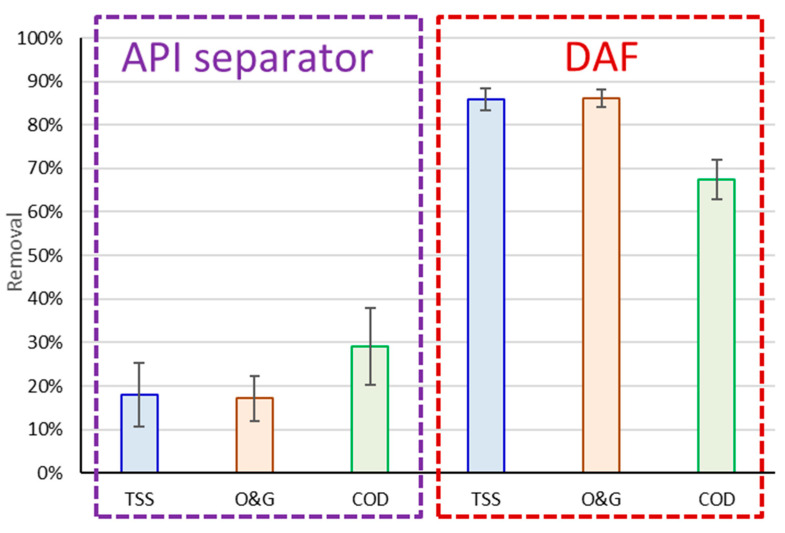
% removals for three main pollutants, mean and standard deviation.

**Table 1 membranes-13-00715-t001:** Published bench-scale MBR test indicating removals and fouling at different O&G levels.

	COD		O&G		Flux		
Membrane	In, mg·L^−1^	Removed	In, mg·L^−1^	Removed	LMH	O&G Impacts on Fouling	Refs.
Ceram. iFS	840–1960	96–99%	-	-	10–40	Not studied	[4]
PVDF iFS	2600	90–92%	1750	96%	2	Sustainable operation at flux imposed	[5]
PVDF iFS	600–7850	95 ± 4%	17–260	94–99%	13	Increasing O&G level from 83 to 260 mg/L accelerated fouling	[6]
PVDF iFS	2600	82–95%	500	85–94%	1.9	Accumulation of O&G in mixed liquor	[7]
PVDF iMT	2000 ± 100	95–96%	5–500	-	4	Dramatic and immediate permeability loss between 200 and 500 mg/L	[8]
PVDF iHF *	783	93%	89	95%	-	Not studied	[12]

* Pilot-scale; Ceram—ceramic membrane; PVDF—poly(vinylidene difluoride); iFS—immersed flat sheet; and iHF—immersed hollow fibre.

**Table 2 membranes-13-00715-t002:** Unit process dimensions.

	Length	Width	Depth	Volume	Footprint	Res Time	SECmax
Unit Process	m	m	m	m^3^	m^2^	h, mins	kWh/m^3^
API unit	34	6	2.8	571.2	95.2	7.6	Low
Neutralisation tank	-	-	-	1500	-	20	Negl.
Equalisation tank	-	-	-	1800	-	24	Negl.
DAF							
DAF unit, total	9.5	3.2	2	60.8	30.4	49	0.127 *
DAF unit, flotation	6.5	2.3	1.7	25.4	15.0	20	-
Coagulation basin	1.6	1.5	2	4.8	2.4	3.8	-
Flocculation basin	1.6	1.7	2	5.4	2.7	4.4	-

* Refers to total installed power (12.7 kW) per rated maximum flow rate (100 m^3^/h).

**Table 3 membranes-13-00715-t003:** Published CAPEX estimates encompassing a range of flow capacities.

Topic	*Q_P_*, m^3^/d	Approach	Refs.
Package plant MBRs	1–10	Summing cost of individual components	[25]
Small MBRs	100–2500	Summing cost of individual components	[26]
Municipal MBRs in Japan	240–6000	Data captured from existing installations	[22]
Municipal MBRs in Spain	300–35,000	Data captured from existing installations	[23]
Municipal MBRs, global	240–38,000	Data captured from existing installations	[21]
Municipal MBRs in US	190–38,000	CAD software, *CAPDETWorks*	[24]
Municipal MBRs in China	>10,000	Cost benefit of MBR retrofit	[27]

*Q_P_*—flow capacity.

**Table 4 membranes-13-00715-t004:** Assumed/calculated values for MBR OPEX determination, adapted from [21].

Parameter	Symbol	Value(s): Base, Range
Oxygen content of air, %	*C’_A_*	21%
SAD, membrane scouring, Nm^3^·m^−2^·h^−1^	*SAD_m_* ^a^	0.225
Mass consumption of oxygen, g·m^−3^	*D_O_* _2_	*Calculated*
SEC, biological aeration, kWh·m^−3^	*E_A,bio_*	*Calculated*
SEC, membrane permeation, kWh·m^−3^	*E_L,m_* ^b^	0.008
SEC, sludge pumping, kWh·m^−3^	*E_L,sludge_* ^c^	0.0161*R*
SEC, membrane air scouring (air), kWh·Nm^−3^	*E’_A,m_* ^d^	*Calculated*
SEC, membrane aeration (permeate), kWh·m^−3^	*E_A,m_* ^e^	*Calculated*
Depth of aerator in process, membrane tank, m	*h*	5, 3.5
Permeate net flux, L·m^−2^·h^−1^ (LMH)	*J*	12 ^f^
Blower coefficient	*k*	*Calculated*
Chemicals consumption costs, USD·m^−3^ permeate	*L_Chem_*	0.02 ^g^
Electricity supply cost, USD·kWh^−1^	*L_E_*	0.2
Membrane cost, USD·m^−2^ membrane area	*L_M_*	85
Operating cost, USD·m^−3^ permeate	*L_O_*	*Calculated*
Oxygen transfer efficiency per unit depth, m^−1^	*OTE*	0.045
Permeate flow rate, m^3^·d^−1^	*Q_P_*	1800
Membrane-biological process tank recycle ratio	*R*	5
Change in COD concentration, g·m^−3^	Δ*S_COD_*	*Experimentally measured*
Change in TKN concentration, g·m^−3^	Δ*S_TKN_*	40
Membrane life, h	*t_MBR_*	70,080
MLSS concn, process, membrane tanks, kg·m^−3^	*X, X_m_*	8, 10
Observed sludge yield, kgSS·kgCOD^−1^	*Y_obs_*	0.35
Mass transfer correction factors	*β*, *γ*	0.95, 0.89
Biomass COD content, kg·kgSS^−1^	*λ_COD_*	1.1
Total pumping electrical energy efficiency	*ε_tot_*	65%
Air density, kg·m^−3^	*ρ_A_*	1.23
Conversion (permeate/feed flow)	*Θ_MBR_*	95%

SAD specific aeration demand; SEC specific energy consumption; DS dry solids; ^a^ air flow rate/membrane area for air scour; ^b^ pump absorbed power/permeate flow rate; ^c^ pump absorbed power/sludge flow rate; ^d^ blower absorbed power/air flow rate; ^e^ blower absorbed power/permeate flow rate; ^f^ from Table 5 of [3]; and ^g^ based on sodium hypochlorite and citric acid costs and usage [14]. Calculation of parameters is as indicated in Table 5.

**Table 5 membranes-13-00715-t005:** MBR OPEX-related equations [21].

Parameter	Symbol	Equation	
*Membrane*			
SEC membrane, kWh·m^−3^	*E_m_*	1000*E’_A,m_SAD_m_*/*J* + *E_L,sludge,i_R_i_* + *E_L,m,i_*	
*Process biology (assuming MLE process denitrification)*			
Oxygen demand, kg·m^−3^	*D_O_* _2_	Δ*S_COD_* (1 − *λ_COD_Y_obs_* − 1.71*λ_TKN_Y_obs_*) + 1.71Δ*S_TKN_*	
SAD, Nm^3^·m^−2^·h^−1^	*SAD_bio_*	*D_O2_*/(*ρ_A_ C’_A_* SOTE *y α β γ*)	=*Q_A,bio_*/*Q_F_*
*α* factor	*α*	e^−0.084*X*^	
SEC, aeration, kWh·Nm^−3^	*E’_A_*	*k* ((0.0943*h* + 1)^0.283^ − 1)/*ε_tot_*	where *k* = 0.107 kWh·Nm^−3^
SEC, permeate, kWh·m^−3^	*E_A,bio_*	*E’_A_ SAD_bio_*	
*Overall OPEX*			
Cost m^−3^ permeate, USD·m^−3^	*L_O_*	*L_E_* (*E_m_* + *E_A,bio_*) + *L_M_*/(*J t*) + *L_Chem_*	

**Table 6 membranes-13-00715-t006:** Estimated CAPEX [29].

Stage	Quoted Cost	Ref, *Q_ref_* or *A_ref_*		Expon.	Coeff.	Corrected Cost		*L_C_*
	2007 USD	L/s or m^2^	m^3^/d	*r*		2007 USD	2023 USD	USD per m^3^/d
API	USD 190,000	12	1037	0.84	1.589	USD 301,994	USD 459,031	USD 255
DAF	USD 1,225,000	50	-	0.48	0.788	USD 964,739	USD 1,466,403	USD 815

Coefficient = (*Q*/*Q_ref_*)*^r^* or (*A*/*A_ref_*)*^r^* where *r* = exponent (Equation (3)); *Q* = 1800 m^3^/d, *A* = 30.4 m^2^ (Table 2); CEPCI ratio, 2023 vs. 2007 = 1.52; API and DAF size adjusted on basis of flow and surface area, respectively.

**Table 7 membranes-13-00715-t007:** MBR energy consumption.

Parameter	Units	Value	Parameter	Units	Value
*Membrane*			*Process biology*		
*E’_A,m_*	kWh·Nm^−3^	0.0138	*E’_A,bio_*	kWh·Nm^−3^	0.0192 ^1^
*SAD_p_*	Nm^3^·m^−3^ permeate	18.8	*D_O2_*	kg·m^−3^	533 ^1^
*E_A,m_*	kWh·m^−3^ permeate	0.259	*SAD_bio_*	Nm^3^·m^−3^ permeate	17.3
*E_m, total_* ^2^	kWh·m^−3^ permeate	0.348	*E_A,bio_*	kWh·m^−3^ permeate	0.509

^1^ Based on a Δ*S_COD_* of 833 mg/L, as indicated by Figure 3 and assuming 90% removal by the MBR; ^2^ including *E_L,m_* and *E_L,sludge_* (Table 4).

**Table 8 membranes-13-00715-t008:** Summary of key performance metrics.

Parameter	API	DAF	MBR
Ave COD load, kg·m^−3^	577	1971	833
SEC, kWh·m^−3^	0.016 ^1^	0.075	0.857
SEC/load, Wh·kgCOD^−1^	0.028	0.038	1.03
NPV, USDm	2.4–2.5		14.4
USDkNPV·kgCOD^−1^	0.9–0.94		17

^1^ Conservatively equated to the sludge pumping cost of the MBR.

## Data Availability

Not applicable.

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
