# Peer review of "The Cost Benefit of Refinery Effluent Pretreatment Upstream of Membrane Bioreactors"

_membranes, 2023, doi:10.3390/membranes13080715_

Round 1

Reviewer 1 Report

1. Few abbreviation such as API ant etc are not mentioned in full when it appear firstly in abstract.

2. Did you carried out laboratory study for the membrane filtration system using the pretreated wastewater? If not, how confirm you are for the reported performance to be accurate enough?

Author Response

Reviewer 1 comments

  1. Few abbreviation such as API ant etc are not mentioned in full when it appear firstly in abstract.
  2. Did you carried out laboratory study for the membrane filtration system using the pretreated wastewater? If not, how confirm you are for the reported performance to be accurate enough

Response

  1. Abbreviations are now defined when first introduced in the abstract as well as in the main text.
  2. The objective of the paper is to produce a comparative cost analysis quantifying the relative efficacies of the physicochemical pretreatment and MBR-based biological treatment of oil refinery effluent. Although the removal of the oil refinery contaminants by the MBR is obviously a pivotal part of the paper, it is not a novel aspect of the research. As stated in this paper (Lines 67-68), and identified specifically in our previous review article (Table 5 of [3]), several studies have demonstrated the efficacy of MBRs for this and related petroleum effluent treatment duties. The authors could thus be reasonably confident that the parameters assumed for the cost analysis in the current paper (see Table 4) are sufficiently accurate for calculating the net present value (NPV). Against this, the available information published on pretreatment for this duty is much more scant. Studies relating to refinery effluent treatment specifically have been based predominantly in coagulation optimisation at bench or small pilot scale [17-18] and have not addressed cost benefit – unlike the MBR stage where some cost analyses have been presented (e.g. [21]). The primary focus of this study was therefore on the pretreatment stage, and specifically its relative cost and efficacy with respect to both energy consumption and pollutant (COD/O&G) removal.

See also attached file for full response to all reviewers' comments.

Reviewer 2 Report

Manuscript Number: membranes- 2535228

The cost benefit of refinery effluent pretreatment upstream of membrane bioreactors.

GENERAL COMMENTS

This article calculates the costs of treating effluents from the oil industry using membrane bioreactors. Analyze the costs of the entire treatment process by comparing various pre-treatments. I found it a rather interesting article that delves into the total cost of treating an industrial effluent and whose content is not easy to find in the literature in such detail. It is written very clearly.

I think the work should be accept in present form.

I just have a few questions/recommendations for authors.

- In figure 4 the title of the y-axis is missing.

- Line 213-214: Is there any explanation for the fact that the elimination of COD by the DAF is lower than that found by other authors?

Author Response

Reviewer 2 comments

This article calculates the costs of treating effluents from the oil industry using membrane bioreactors. Analyze the costs of the entire treatment process by comparing various pre-treatments. I found it a rather interesting article that delves into the total cost of treating an industrial effluent and whose content is not easy to find in the literature in such detail. It is written very clearly.

I think the work should be accept in present form.

I just have a few questions/recommendations for authors.

  1. In figure 4 the title of the y-axis is missing.
  2. Line 213-214: Is there any explanation for the fact that the elimination of COD by the DAF is lower than that found by other authors?

Response by authors

  1. The missing axis title has been added to the figure in the revised manuscript.
  2. The difference appears to relate mainly to coagulant dose, as explained in the manuscript Lines 218-222: Much lower COD removals of around 68% - comparable with those values employed in the current study – were obtained at a lower coagulant dose of around 25 mg.L-1 based on a previous bench-scale test [18]. The maximum coagulant dose applied in the current study was 40 mg/L.

See also attached file for full response to all referees' comments.

Reviewer 3 Report

1: Kindly write full abbreviation of COD and AP-I separator (especially in abstract)

2: Keywords: write them in alphabetic order

According to literature, there are several studies relevant to biorefinery with better results than the current study such as:

3:Question:  Janson et al. (2015)  used MBR for the biotreatment of hydrate-inhibitor containing produced waters and obtained more than  90% of effluent removal. How do you differentiate this study from your current research, in terms of novelty?

4. Question: Atia et al. (2019) found that the efficiency of the filtration can reach more than 90% if it is improved by adding coagulants before the filtration process which seems more cost effective as compared to your study. Discuss the cost effectiveness?

5. Suggestion: Please compare the results of membrane processes for treatment technologies such as UF, MF, NF, reverse and forward osmosis with membrane-bioreactors (MBRs).

6: Kindly highlight the novelty part.

Author Response

Reviewer 3 comments

1: Kindly write full abbreviation of COD and AP-I separator (especially in abstract)

2: Keywords: write them in alphabetic order

According to literature, there are several studies relevant to biorefinery with better results than the current study such as:

3: Question:  Janson et al. (2015)  used MBR for the biotreatment of hydrate-inhibitor containing produced waters and obtained more than  90% of effluent removal. How do you differentiate this study from your current research, in terms of novelty?

4: Question:Atia et al. (2019) found that the efficiency of the filtration can reach more than 90% if it is improved by adding coagulants before the filtration process which seems more cost effective as compared to your study. Discuss the cost effectiveness?

5: Suggestion:Please compare the results of membrane processes for treatment technologies such as UF, MF, NF, reverse and forward osmosis with membrane-bioreactors (MBRs).

6: Kindly highlight the novelty part.

Response by authors

  1. Abbreviations are now defined when first introduced in the abstract as well as in the main text.
  2. Keywords are now listed alphabetically.
  3. Produced water (PW) has a substantially different composition to refinery effluent, the subject of the current study. In the Janson et al (2015) study, 1.5% each of kinetic hydrate inhibitor and monoethylene glycol were added to a PW sample, differentiating the composition significantly from refinery effluent, and the microbiology of the MBR (as shown in Fig 5(a) of the Janson et al paper) was also highly unusual. In the current study, a comparative cost analysis, the assumed performance of the MBR was based ostensibly on data from full and pilot scale data for refinery effluent, as summarised in Table 5 of a previous review article [3]. The key parameter impacting on the cost is the flux, where a value of 12 LMH has been assumed based on the available published information. This important assumption is now referenced in both the footnote to Table 4 and the second paragraph in Section 3.2 (Lines 286-287) of the revised manuscript.
  4. The only papers the authors were able to find fitting the descriptor “Atia et al, 2019” refer to PW treatment. The effectiveness of the clarification stage is nonetheless determined by largely by the coagulation chemistry. It has been acknowledged in the third paragraph of the Results and Discussion section (Lines 214-217) that far higher removals of 92-98% O&G, COD and TSS have been reported for a fully optimised DAF challenged with oil refinery effluent [17]. However, achieving these removals demanded a dose of 100 mg/L coagulant – significantly higher than the dose used at full-scale in the current study. Much lower COD removals of around 68% – comparable with those values employed in the current study – have been reported when using a lower coagulant dose of around 25 mg/L [18], which is comparable to the 25-40 mg/L dose range for the current study.
  5. The relative performance of membrane processes generally can be found in a number of text and reference books on the subject. The application and performance of membrane processes for oil refinery effluent treatment specifically has been reviewed by Munirasu et al (2016). A comparison of abiotic membrane filtration and MBRs has been more recently reviewed, and cited in the current paper [3]. The application of dense membrane processes (RO, ED and NF) to oil refinery effluents is limited to a few instances of water reuse downstream of an MBR. Forward osmosis does not appear to have been implemented beyond the demonstration scale, according to Awad et al (2019).
  6. The novelty of the paper is in the comparative cost analysis quantifying the relative efficacies of the physicochemical pretreatment and MBR-based biological treatment of oil refinery effluent. This is now emphasised in Line 98 at the end of the Introduction section in the revised manuscript. The text in the Conclusions (Lines 348-349) has similarly been changed.

References

Awad, A. M., Jalab, R., Minier-Matar, J., Adham, S., Nasser, M. S., Judd, S. J. (2019). The status of forward osmosis technology implementation. Desalination, 461 10-21.

Janson, A., Santos, A., Hussain, A., Judd, S., and Adham, S. (2015). Biotreatment of hydrate inhibitor-containing produced waters at low pH, SPE Journal 20 (6), 1254-1260.

Munirasu, S., Haija, M.A., Banat, F. (2016). Use of membrane technology for oil field and refinery produced water treatment - A review. Process Safety and Environmental Protection 100, 183-202.

Round 2

Reviewer 3 Report

Dear editor,

Greetings

The manuscript has been improved significantly, all the suggestions, questions and recomendations have been in-cooperated accordingly. My decision is to accept the manuscript in its current form.

Thank you